# Factors associated with seclusion and restraint on admission to forensic psychiatric hospitals: A 10-year retrospective study

Elke Ham[1], Soyeon Kim[1,2], N. Zoe Hilton ![ORCID][1,3,4]*

1 Waypoint Research Institute, Waypoint Centre for Mental Health Care, Penetanguishene, Ontario, Canada, 2 Psychiatry and Behavioral Neurosciences, McMaster University, Hamilton, Ontario, Canada, 3 Temerty Faculty of Medicine, University of Toronto, Toronto, Ontario, Canada, 4 Dalla Lana School of Public Health, University of Toronto, Toronto, Ontario, Canada

* zoe.hilton@utoronto.ca

## Abstract

The use of coercive measures such as seclusion and restraint in forensic mental healthcare settings is widespread but controversial. Efforts to reduce these measures require knowledge of patient-related risk factors. The present study aimed to identify and confirm factors related to seclusion and restraint that can be assessed upon admission among men and women admitted to forensic hospitals in Ontario, Canada. We included cross-sectional Ontario Mental Health Reporting System admission data for adult patients admitted to 10 forensic psychiatric hospitals between April 1, 2013, and March 31, 2023. We determined patient demographic, administrative, and clinical characteristics associated with seclusion and physical and manual restraint episodes during the first three days of admission. We conducted logistic Generalized Linear mixed Models (GLMM) to examine the association between the independent variables and restraint and seclusion while accounting for variability across facilities. Of 7635 patients, 30.2% (n = 2302) were secluded, and 3.7% (n = 286) were restrained within their first three days of admission. Secluded patients were more likely to be young adults, male, and scored higher on violence and aggression measures. Being admitted due to fitness-related reasons, lack of insight, medication non-adherence, higher scores on the mania scale and cognitive impairment further contributed to the higher odds of being secluded, whereas neurocognitive disorder diagnosis and elopement behavior were protective factors. Restrained patients were also more likely to be young adults, have a diagnosis of mood or anxiety, neurodevelopmental or personality disorder, and scored higher on violence and aggression measures. Fitness-related status, medication non-adherence, and cognitive impairment further contributed to this model of restraint. Indigenous self-identification and immigration status were not significant contributors to either model. Clinicians can assess indicators associated with seclusion and restraint when forensic patients are admitted to

**Data availability statement:** Data cannot be shared publicly because of a data use agreement. Data are available from the Canadian Institute for Health Information (CIHI) through the Ontario Mental Health Reporting System (OMHRS) to researchers who meet the criteria for access to confidential data, as outlined by CIHI: https://www.cihi.ca/en/ontario-mental-health-reporting-system-metadata.

**Funding:** The author(s) received no specific funding for this work.

**Competing interests:** The authors have declared that no competing interests exist.

forensic hospitals or during the first three days of their stay, enabling effective targeting of those needs to reduce the use of coercive measures.

## Introduction

Coercive measures used in healthcare involve interventions implemented against a patient's wishes, such as isolation, physical or mechanical restraint, and enforced medication. Coercive measures are frequently employed in psychiatric settings to help manage severe patient behaviors that pose a threat to the patient or others [1]. A meta-analysis of 77 adult mental health settings across five continents found that the pooled prevalence of physical restraint, seclusion, and chemical restraint was 14%, 16%, and 26%, respectively [2]. In forensic hospitals [secure hospitals providing psychiatric care for justice-involved individuals], seclusion is a particularly prevalent means of controlling aggressive or violent behavior [3–5]. Hui and colleagues reported in their review that between 28% and 44% of patients in forensic psychiatric settings internationally were secluded during their hospitalization [6], and an Ontario study reported that 18% of forensic in-patients were secluded within the previous three days of an assessment [7].

Coercive measures raise significant legal [8], ethical [9], and human rights concerns [10]. Patients perceive seclusion as emotionally harmful and punitive [11] and may incur physical harm [12,13], including fatalities [14]. Staff engaging in seclusion and restraint may experience physical and psychological injuries [15,16], frustration and burnout [17–19]. The use of seclusion and restraint can have negative economic impacts on organizations [20], whereas decreased use can reduce staffing costs, sick time, turnover, and workers' compensation [21]. Understandably, seclusion and restraint are among the most heavily regulated practices in psychiatry [22] and may only be used to prevent immediate harm to a patient or others, and after all less restrictive interventions have been tried [12,23,24].

Despite considerable efforts to reduce seclusion and restraint [25], there is limited evidence of successful reductions [26–28] and even some reports of increased use [4]. Critics have argued for enhanced regulation of coercive practices [29]. Yet, policies may fail because of a belief that seclusion is necessary for the safety of staff and patients [30] despite evidence to the contrary [31]. Failure to reduce the seclusion of restraint has led to recommendations for improved clinical techniques and teamwork [32]. Providing individualized care is an important component of clinical efforts to reduce the use of seclusion and restraint [33,34]. The effectiveness of such interventions can be optimized by identifying the risk of individuals being subjected to coercive measures and designing and implementing alternative strategies for those most at risk. A review of 49 studies of physical restraint in psychiatric hospitals found that the most often reported risk factors were male, young adult age, foreign ethnicity, diagnosis of schizophrenia, involuntary admission, aggression, escape attempts, and the presence of male staff [35]. Importantly, the use of coercive measures is most common in the first few days of treatment [36]; in this time period, clinical

parameters such as manic or psychotic episodes are important risk factors [36]. Involuntary admission [36–39] and elopement attempts [35,40] also predict the frequency or use of restraints. The association of involuntary status with the use of coercive measures aligns with increased risk among forensic admissions. However, comprehensive investigations of risk factors for coercive measures in forensic psychiatric institutions are sparse.

## Seclusion and restraint in adult forensic psychiatric services

In Canada, individuals may be detained in a forensic mental health hospital for two main reasons: 1. For short-term evaluation and treatment to assess their fitness to stand trial or determine if they had a mental disorder at the time of the alleged offense, possibly exempting them from criminal responsibility; and 2. For long-term rehabilitation and reintegration after the court finds them unfit to stand trial (UST) or not criminally responsible due to a mental disorder (NCRMD) [41]. While some forensic patients may be admitted to units with minimum security (i.e., open or locked door, limited unaccompanied time off unit), many begin their forensic hospitalization under medium security (i.e., locked door with restricted time off unit) or high security (i.e., perimeter security, locked door, restricted movement within the facility).

Among adult patients in forensic settings, there is established evidence that younger adults are more likely to be secluded [6,42–44]. Some clinical and behavioral characteristics are also consistent risk factors, including violent and aggressive behavior [6,43,45] and psychiatric diagnoses of schizophrenia and other psychotic disorders [6,42–44], personality disorders [43,44], and substance use disorder [42]. Less extensive research indicates an association with mania, cognitive impairment, lack of insight, and medication non-compliance [7]. There is inconsistent evidence regarding gender differences [6,7,43]. Further, some additional administrative and behavioral characteristics that are risk factors for coercive measures in general psychiatric settings have not yet been examined in forensic settings, including elopement attempts, legal status (e.g., NCRMD vs. UST), and recent hospital admission, which is a time of intensive assessment in forensic hospitals and a prime opportunity to assess and mitigate risk factors.

Most research on coercive measures in forensic settings concerns seclusion, not restraint [6], and relies on bivariate designs rather than multivariable analyses that can tease out the strongest risk factors to inform the most effective interventions. Moreover, no previous studies have analyzed a broad spectrum of clinical, demographic, and administrative factors that can be used in the crucial first few days of forensic admission to identify at-risk individuals for preventative interventions. In addition, there have been calls for increased attention to the experience of racialized and Indigenous individuals in forensic healthcare systems [46–48] in light of reported racial bias within correctional and forensic systems whereby Indigenous people [49,50] and racial minorities [6,43,49] may be disproportionately subjected to seclusion and restraint.

The present study aimed to identify and validate factors associated with seclusion and restraint that can be assessed upon admission, using a large sample of men and women admitted to forensic hospitals in Ontario, Canada. Specifically, we examined the relative contributions of previously established demographic and clinical risk factors, as well as inconclusive or understudied factors, in predicting the use of coercive measures within the first three days of admission. We hypothesized that younger age, diagnoses of psychotic disorder, substance use disorder, and personality disorder, along with violent and aggressive behavior, would be significant predictors. Additionally, we examined the role of additional demographic (male sex, immigration background, Indigeneity), administrative (legal status), and clinical factors (elopement, lack of insight, medication non-adherence, mania, and cognitive impairment), hypothesizing that these would also contribute to the likelihood of seclusion and restraint in forensic hospitals.

## Method

This study was part of a project investigating changes in the prevalence and correlates of mental health disorders in psychiatric in-patients in Ontario, Canada. This study received approval from the authors' institutional research ethics board with a waiver of patient consent based on the Tri-Council Policy Guidelines for waiver of consent (HPRA#19.12.03).

## Data sources

Data were extracted from the Ontario Mental Health Reporting System on the Resident Assessment Instrument – Mental Health (RAI-MH) [51], which evaluates demographics, mental and physical health symptoms, substance use, behaviors, service utilization, and various functioning domains [52,53] on all adult psychiatric in-patient hospitalizations in Ontario. RAI-MH is a standardized clinical tool used to regularly assess psychiatric in-patients, with adequate reported inter-rater reliability (> 80% average percent agreement across all items) [53] and convergent validity for clinical scales [54]. We accessed the anonymized data on 13.10.2023 for research purposes, and information that could identify individual patients during or after data collection was not available or accessed. The variables used in the study were mandatory reporting requirements for Ontario hospitals and must be completed within 72 hours of admission to a mental health bed. A complete dataset with no missing information was obtained, ensuring that missingness did not pose a concern for data analysis.

## Sample

The study included patients for whom full admission RAI-MH assessments (minimum 3-day admission) were completed from April 1, 2013, to March 31, 2023. We accessed data from 9001 forensic patient admissions to the 10 Ontario forensic psychiatric institutions designated facilities under part XX.1 of the Criminal Code of Canada [55]. We identified duplicate admissions based on the encrypted Ontario health identification number, and for cases with multiple admissions, we used the first admission per fiscal year studied. Patients without an Ontario health identification number are assigned a non-unique number and can include out-of-province residents or recent newcomers; 433 admissions fell under this category, and to help identify unique cases, we selected the first admission per fiscal year by sex, Indigenous-self identification, age, and hospital. We first deleted 1313 duplicates based on the unique identifier and then deleted 53 duplicates of the 433 cases with the same ID. We retained 7635 cases for analysis.

## Study variables

Outcome variables included any seclusion and any physical or mechanical restraint documented during the first three days of admission. Seclusion was defined as present when a patient was locked in a room that confined them and from which they cannot leave freely, including being placed in a special seclusion room ($n = 1323$) or confined to their room ($n = 1457$) [56]. Restraint was defined as either mechanical restraint, whereby a patient was being restrained in bed or through wrist restraints only ($n = 193$), or physical restraint, whereby the patient was physically held to restrict their movement for a brief period to restore calm to the individual ($n = 167$)]; although mechanical and physical holding are different restraint strategies, they had similar durations (i.e., modal use < daily) and due to the relatively low proportion of the sample, it was necessary to combine these cases to capture all restraint, similar to Hui and colleagues findings [57]. The study included 10 forensic in-patient facilities in Ontario, which comprised one 160-bed maximum secure hospital, as well as facilities featuring 470 medium secure beds and 271 minimum secure beds.

Demographic variables included age in years at the time of admission, sex (male, female, other), whether the patient had immigrated to Canada, and Indigenous self-identification (based on patients' self-identification as First Nations, Metis, or Inuit) [52].

Our administrative variable included forensic status at assessment. This RAI-MH variable consisted of nine categories, which we combined into three categories: 1 = Fitness-related status (fitness assessment, warrant of committal-unfit, treatment order, keep fit order and inter-hospital transfer-unfit); 2 = Not Criminally Responsible (NCR) – related status (NCR assessment, warrant of committal-NCR and inter-hospital transfer-NCR); and 3 = Other (e.g., court-ordered disposition recommendations).

Clinical variables included psychiatric diagnoses, insight into mental illness and medication compliance indicators, and several scales measuring symptoms and behavior [52]. Up to three diagnoses were documented using DSM-IV (n = 2,501)

or DSM-5 (n = 5,134). We recoded all diagnoses to indicate whether each individual had a diagnosis within each of six categories: 1 = schizophrenia spectrum and other psychotic disorders, 2 = substance use disorders, 3 = mood or anxiety disorders, 4 = neurocognitive disorders, 5 = personality disorders, 6 = neurodevelopmental disorders.

The RAI-MH includes a measure to assess a patient's awareness of their mental health issues and contributing factors [52]. This item was recorded as full, limited, or no insight, which we categorized as 0 = full or limited insight and 1 = no insight. History of Medication Adherence is a measure to determine if the patient took their medication as prescribed in the month prior to admission. The item was coded as 0 = at least 80% adherent or no medication prescribed, and 1 = less than 80% adherent or adherence unknown. The RAI-MH includes various behavioral symptom measures, including elopement attempts/threats. Elopement occurs when an individual attempts to leave or leaves the unit without staff knowledge or formal discharge (unauthorized leave). Elopement attempts/threats were recorded from (0) not exhibited in the last 3 days to (3) symptoms that occurred every day during the last three-day period [40,52]. We coded this item as 0 = not present and 1 = present.

We included four clinical scales derived from RAI-MH items [51,53,58]. The Aggressive Behavior Scale (ABS) is composed of four items (verbal abuse, physical abuse, socially inappropriate/disruptive behavior, resistance to care), each scored from 0 (not exhibited) to 3 (exhibited daily) with a total score ranging from 0 to 12 [59,60]. The Violence Sum (VS) sums three items (violent acts, intimidation, ideation), each scored from 0 (never) to 5 (in the last three days), and the total score ranges from 0–15 [58]. The Mania scale measures the frequency of seven symptoms (inflated self-worth, hyperarousal, irritability, pressured speech, labile affect, sleep problems due to hypomania, increased sociability/hypersexuality) with a total score ranging from 0–20 [51]. The Cognitive Performance Scale includes four items that assess short-term memory and recall, daily decision-making, communication abilities, and self-performance in eating. Scores on this decision tree algorithm-based scale range from 0 (indicating intact cognitive function) to 6 (indicating very severe impairment) [53].

## Data analysis

We calculated descriptive statistics for admission, seclusion, and restraint frequencies, as well as for sociodemographic, administrative, and clinical variables. For continuous variables, including age, the mean (M) and standard deviation (SD) are reported. For length of stay, the mean (M), standard deviation (SD), median, and interquartile range (IQR) were provided. Categorical variables were summarized using the number (N) and percentage (%). We conducted unadjusted binary logistic regression analyses with independent variables to examine individual association with seclusion and restraint. To account for the clustering of responses within facilities/hospitals, we employed logistic Generalized Linear Mixed Model (GLMM). This approach allowed us to estimate the strength of the association between coercive measures (seclusion and restraint) and patient-level factors, including sociodemographic, administrative, and clinical factors, accounting for clustering within hospitals. We analyzed seclusion and restraint separately. All statistical analyses were completed using SPSS Version 29.

## Results

7635 admissions met our criteria. The mean length of stay was 235.06 (SD = 451.64) days, median 57 days, and IQR (29, 179). The mean age at admission was 37.69 years (SD = 12.76), with a median age of 35 and IQR (28, 46). A majority were identified as male (n = 6398, 84%) compared with female (n = 1231, 16%) or other sex (n = 6, 0.1%). 502 (7%) self-identified as Indigenous, and less than 20% (n = 1448) had an immigration background. More than 2/3 of patients had a primary diagnosis of psychotic disorder (n = 5277, 69%), followed by mood and anxiety disorder (n = 855, 11%), and almost half had a substance use disorder diagnosis (n = 3688, 48%). One-quarter of patients had no insight (n = 2042, 27%), and 1/3 were less than 80% medication adherent (n = 2887, 38%). 5% of patients (n = 387) exhibited elopement behavior. Almost half (n = 3720, 49%) were admitted for either a fitness assessment (n = 1667, 22%) or an NCR

assessment (n = 2053, 27%). 1 in 5 were admitted on an NCR warrant of committal or inter-hospital transfer (n = 1592, 21%), and another 1 in 5 for a treatment order or keep fit order (n = 1648, 22%). A small proportion of patients was admitted (or interhospital transfer) after being found unfit (193, 2%). 482 admissions were for other reasons (6%). Additional details are reported in S1 Table.

Yearly admissions ranged from a low of 539 patients in 2020 to a high of 866 in 2016. The average 10-year seclusion rate was 30.2%, with a low of 24.9% in 2013 and a high of 49.7% in 2020; The average 10-year restraint rate was 3.7% with a low of 2.7% in 2016 and a high of 5.4% in 2014; see Fig 1.

Seclusion rates differed between facilities, ranging from 13.6% to 64.2%, with a median seclusion rate of 22.6% (IQR: 20.6, 35.4). Restraint rates also differed between facilities, ranging from 1.1% to 9.1%, with the median of 3.6% (IQR: 2.3, 5.1; see Fig 2).

### Risk factors for seclusion

Table 1 presents the comparison between secluded and non-secluded patients and the results of unadjusted binary logistic regression analyses examining the association between individual sociodemographic, administrative, and clinical factors and seclusion as the outcome variable. Secluded patients were significantly younger than their non-secluded counterparts. Unadjusted odds ratios showed that seclusion was significantly associated with younger male patients and those with a diagnosis of psychotic disorder, exhibiting lack of insight, being less than 80% medication compliant in the previous month, showing elopement behaviors, and with higher scores on the ABS, Violence Sum and Mania scales and higher cognitive impairment. Patients whose forensic status was fitness-related were almost three times more likely to be secluded than those whose status was NCR-related. Substance use disorder or neurocognitive disorder diagnosis and immigration background were protective factors in these bivariate analyses. Indigenous self-identification did not significantly affect seclusion outcomes.

The overall logistic GLMM model for seclusion (Table 2) was significant, $F(19, 7609) = 62.140$, $p < .001$, indicating that the predictors significantly explained variability in seclusion after accounting for hospital-level clustering effects. As expected, age and our measures of aggression and violence were significantly associated with higher odds of being secluded,

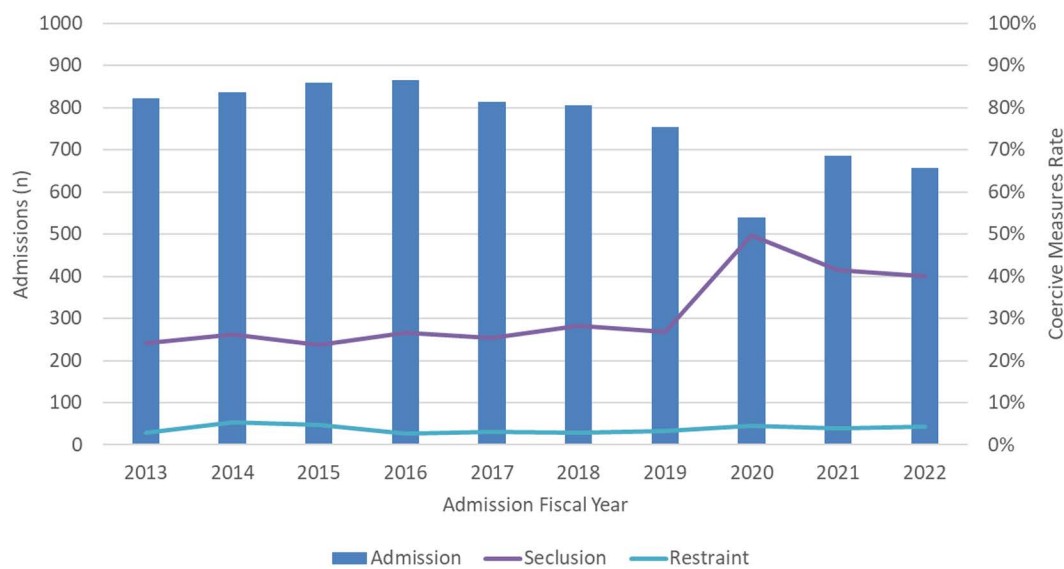

**Fig 1. Coercive measures and admission to Ontario forensic hospitals over a 10 year period.**

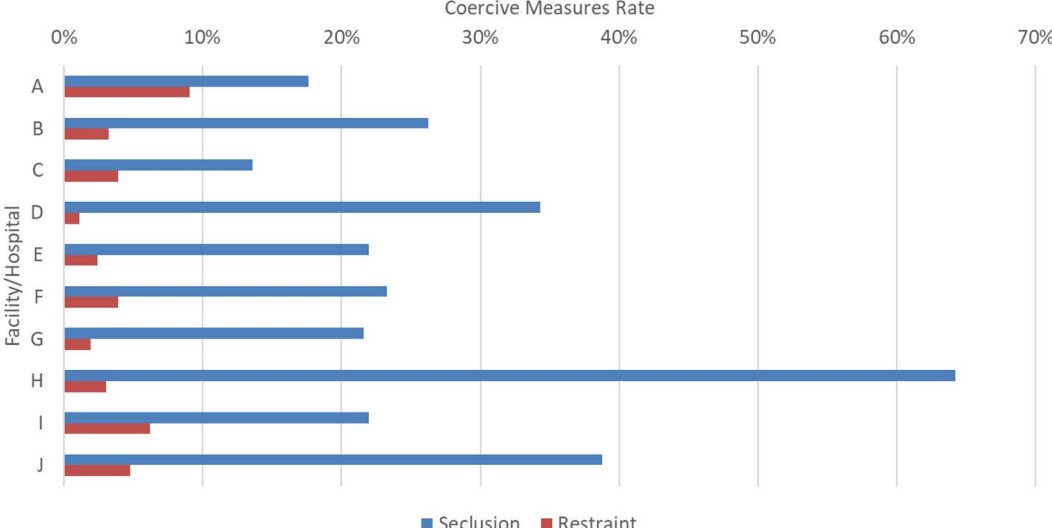

**Fig 2. Coercive measures by Ontario forensic hospitals.**

whereby younger patients and patients exhibiting higher scores on the ABS and Violence Sum scales were more likely secluded, partially supporting our first hypothesis. Regarding diagnoses, a diagnosis of neurocognitive disorder was negatively associated with seclusion, whereas other diagnoses, including psychotic disorder, substance use disorder, and personality disorder, were not significant predictors,

In this logistic GLMM model, being male was significantly associated with higher odds of seclusion; males were 50% more likely to be associated with seclusion than females. Lack of insight, medication non-compliance, higher scores on the Mania scale, and cognitive impairment were also significantly associated with seclusion. Compared to patients admitted for NCR-related reasons, patients admitted for fitness-related reasons were almost twice as likely to be associated with seclusion, whereas elopement was negatively associated. Indigeneity and immigration status were not associated with seclusion in this model.

After adjusting for patient-level predictors, 14% of the variance in seclusion was explained by hospital-level factors (Intraclass Correlation Coefficient [ICC] = 0.14), representing a moderate clustering effect. The fixed effects (patient-level predictors) explained 21% of the variance (Marginal pseudo $R^2 = 0.21$). When accounting for both patient-level predictors and hospital-level clustering effects, the total explained variance increased to 35% (Conditional pseudo $R^2 = 0.35$), indicating that hospital-level factors contributed additional explanatory power. Table 2 presents the associations between patient-level factors and seclusion during the first three days of admission, accounting for the hospital-level clustering effect.

### Risk factors for physical or manual restraint

Table 3 shows results from our unadjusted bivariate regression analyses with restraint as an outcome. Restrained patients were younger compared to non-restrained patients. Unadjusted odds ratios showed that restraint was more likely applied to female patients and patients with a primary diagnosis of neurodevelopmental, personality, or mood and anxiety disorder. In addition, high scores on our clinical measures for aggression, violence, mania, and cognitive impairment were also associated with restraint, as was having no insight into mental health and being less than 80% medication compliant. Having a fitness-related status at admission compared to an NCR-related status increased odds of restraint more than two-fold and exhibiting elopement behaviors increased odds of restraint more than fourfold. Having an immigration background was not a significant factor, nor was Indigenous self-identification.

**Table 1. Demographic, administrative, and clinical characteristics for not secluded and secluded patients and unadjusted associations (OR, 95% CI).**

| Characteristic | No Seclusion (*n*=5333, 69.8%) | Seclusion (*n*=2302, 30.2%) | Unadjusted binary logistic regression analysis of seclusion | | |
| --- | --- | --- | --- | --- | --- |
| | *n* (%) | *n* (%) | *OR* | 95% *CI* Lower | Upper |
| **Age at Admission (years)** M (SD) | 38.6 (13.1) | 35.6 (11.5) | 0.98*** | 0.98 | 0.98 |
| **Sex** | | | | | |
| Female | 921 (17.3) | 310 (13.5) | Ref | | |
| Male | 4412 (82.7) | 1986 (86.3) | 1.34*** | 1.16 | 1.54 |
| **Diagnoses** | | | | | |
| Psychotic Disorder | 3905 (73.2) | 1767 (76.80) | 1.21*** | 1.08 | 1.35 |
| Substance Use Disorder | 2655 (48.9) | 1033 (44.9) | 0.82*** | 0.74 | 0.91 |
| Mood and Anxiety Disorder | 927 (17.4) | 371 (16.1) | 0.91 | 0.80 | 1.04 |
| Neurocognitive Disorder | 259 (4.9) | 59 (2.6) | 0.52*** | 0.39 | 0.69 |
| Personality Disorder | 928 (17.4) | 424 (18.4) | 1.07 | 0.94 | 1.22 |
| Neurodevelopmental Disorder | 365 (6.8) | 136 (5.9) | 0.85 | 0.70 | 1.05 |
| **Immigration Background** | | | | | |
| No immigration background | 4273 (80.1) | 1914 (83.1) | Ref | | |
| Immigration background | 1060 (19.9) | 388 (16.9) | 0.82** | 0.72 | 0.93 |
| **Indigenous Self-Identification** | | | | | |
| Did not identify as Indigenous | 4968 (93.2) | 2165 (94) | Ref | | |
| Identified as Indigenous | 365 (6.8) | 137 (6) | 0.86 | 0.70 | 1.06 |
| **Forensic Status** | | | | | |
| NCR-related status | 2913 (54.6) | 732 (31.8) | Ref | | |
| Fitness-related status | 2034 (38.1) | 1474 (64) | 2.88*** | 2.60 | 3.20 |
| Other reason | 386 (7.2) | 96 (4.2) | 0.99 | 0.78 | 1.25 |
| **Elopement Attempts/Threats** | | | | | |
| Did not display behavior | 5114 (95.9) | 2134 (92.7) | Ref | | |
| Displayed elopement behavior | 219 (4.1) | 168 (7.3) | 1.84*** | 1.49 | 2.26 |
| **Medication Adherence** | | | | | |
| ≥80% medication adherent | 4186 (78.5) | 1407 (61.1) | Ref | | |
| <80% adherent | 1147 (21.5) | 895 (38.9) | 1.97*** | 1.78 | 2.18 |
| **Insight into Mental Illness** | | | | | |
| Full or limited insight | 3578 (67.1) | 1170 (50.8) | Ref | | |
| No insight | 1755 (32.9) | 1132 (49.2) | 2.32*** | 2.09 | 2.58 |
| **Clinical Scales** | | | | | |
| Aggressive Behavior Scale | 1.2 (2.1) | 3.3 (3.4) | 1.32*** | 1.29 | 1.34 |
| Violence Sum | 4.4 (3.2) | 6.4 (3.7) | 1.18*** | 1.16 | 1.20 |
| Mania Scale | 2.5 (3.6) | 5.2 (4.9) | 1.15*** | 1.14 | 1.17 |
| Cognitive Performance Scale | 0.5 (0.9) | 0.8 (1.2) | 1.27*** | 1.21 | 1.33 |

*p <.05, **p <.01, ***p <.001; *OR*, Odd Ratio; *CI*, confidence interval.

**Table 2. Fixed effects for logistic GLMM model predicting seclusion during the first three days of admission with hospital-level clustering effect.**

**Logistic Generalized Linear Mixed Model – GLMM**

| Factor | β | SE | t | p | β2 | 95% CI | |
|---|---|---|---|---|---|---|---|
| | | | | | | Upper | Lower |
| **Intercept** | −2.34 | 0.36 | −6.48 | <.001 | 0.17*** | 0.08 | 0.34 |
| **Age at Admission (years) M** | −0.02 | 0.00 | −7.08 | <.001 | 0.98*** | 0.98 | 0.99 |
| **Sex** | | | | | | | |
| Female | Ref | | | | Ref | | |
| Male | 0.42 | 0.09 | 4.68 | <.001 | 1.52*** | 1.27 | 1.80 |
| **Diagnoses** | | | | | | | |
| No psychotic disorder | | | | | Ref | | |
| Psychotic Disorder | 0.17 | 0.09 | 1.94 | .052 | 1.19 | 1.00 | 1.41 |
| No Substance Use Disorder | | | | | Ref | | |
| Substance Use Disorder | −0.09 | 0.06 | −1.44 | .149 | 0.91 | 0.80 | 1.03 |
| No Mood and Anxiety Disorder | | | | | Ref | | |
| Mood and Anxiety Disorder | 0.15 | 0.10 | 1.55 | .122 | 1.16 | 0.96 | 1.41 |
| No Neurocognitive Disorder | | | | | Ref | | |
| Neurocognitive Disorder | −0.41 | 0.18 | −2.30 | .021 | 0.66* | 0.46 | 0.94 |
| No Personality Disorder | | | | | Ref | | |
| Personality Disorder | 0.02 | 0.08 | 0.22 | .828 | 1.02 | 0.86 | 1.20 |
| No Neurodevelopmental Disorder | | | | | Ref | | |
| Neurodevelopmental Disorder | −0.15 | 0.13 | −1.18 | .239 | 0.86 | 0.66 | 1.11 |
| **Immigration Background** | | | | | | | |
| No immigration background | | | | | Ref | | |
| Has immigration background | −0.12 | 0.08 | −1.46 | .145 | 0.89 | 0.76 | 1.04 |
| **Indigenous Self-Identification** | | | | | | | |
| Did not identify as Indigenous | | | | | Ref | | |
| Identified as Indigenous | −0.17 | 0.13 | −1.28 | .202 | 0.85 | 0.65 | 1.09 |
| **Forensic Status** | | | | | | | |
| NCR-related status | | | | | Ref | | |
| Fitness-related status | .57 | .07 | 8.42 | <.001 | 1.77*** | 1.55 | 2.01 |
| Other reason | .06 | .14 | 0.43 | .667 | 1.06 | 0.81 | 1.39 |
| **Elopement Attempts/Threats** | | | | | | | |
| Did not display behavior | | | | | Ref | | |
| Displayed elopement behavior | −0.28 | 0.14 | −2.03 | .043 | 0.75* | 0.57 | 0.99 |
| **Insight into Mental Illness** | | | | | | | |
| Full or limited insight | | | | | Ref | | |
| No insight | 0.13 | 0.07 | 1.74 | .081 | 1.13 | 0.98 | 1.31 |
| **Medication Adherence** | | | | | | | |
| ≥80% medication adherent | | | | | Ref | | |
| <80% adherent | 0.14 | 0.06 | 2.11 | .035 | 1.15* | 1.01 | 1.30 |
| **Clinical Scales** | | | | | | | |
| Aggressive Behavior Scale | 0.25 | 0.02 | 16.04 | <.001 | 1.28*** | 1.24 | 1.32 |
| Violence Sum | 0.07 | 0.01 | 7.40 | <.001 | 1.08*** | 1.06 | 1.10 |
| Mania Scale | 0.07 | 0.01 | 7.47 | <.001 | 1.07*** | 1.05 | 1.09 |
| Cognitive Performance Scale | 0.10 | 0.03 | 3.11 | .002 | 1.10** | 1.04 | 1.17 |

*p < .05, **p < .001, ***p < .001; β, coefficient; SE, standard error; t, t-test′ β², (exp)coefficient; CI, confidence interval.

**Table 3. Demographic, administrative, and clinical characteristics for not restrained and restrained patients and unadjusted associations (OR, 95% CI).**

| | Not restrained (*n* = 7349, 96.3%) | Restrained (*n* = 286, 3.7%) | | Unadjusted binary logistic regression analysis of restraint | |
|---|---|---|---|---|---|
| **Characteristic** | *n* (%) | *n* (%) | *OR* | 95% CI | |
| | | | | Lower | Upper |
| **Age at Admission (years), M (SD)** | 37.82 (12.77) | 34.46 (12.24) | 0.98*** | 0.97 | 0.99 |
| **Sex** | | | | | |
| Female | 1173 (16) | 58 (20.4) | Ref | | |
| Male | 6171 (84) | 227 (79.6) | .74* | .55 | 1.00 |
| **Primary Diagnosis** | | | | | |
| Schizophrenia/psychotic disorder | 5462 (74.3) | 210 (73.4) | 0.95 | 0.73 | 1.25 |
| Substance use disorder | 3556 (48.4) | 132 (46.2) | 0.91 | 0.72 | 1.16 |
| Mood or anxiety disorder | 1237 (16.8) | 61 (21.3) | 1.34* | 1.00 | 1.79 |
| Neurocognitive disorder | 308 (4.2) | 10 (3.5) | 0.83 | 0.44 | 1.57 |
| Personality disorder | 1285 (17.5) | 67 (23.4) | 1.44* | 1.09 | 1.91 |
| Neurodevelopmental disorder | 470 (6.4) | 31 (10.8) | 1.78** | 1.21 | 2.61 |
| **Immigration Background** | | | | | |
| No immigration background | 5959 (81.1) | 228 (79.7) | Ref | | |
| Immigration background | 1390 (18.9) | 58 (20.3) | 1.09 | 0.81 | 1.46 |
| **Indigenous Self-Identification** | | | | | |
| Did not identify as Indigenous | 6865 (93.4) | 268 (93.7) | Ref | | |
| Identified as Indigenous | 484 (6.6) | 18 (6.3) | 0.95 | 0.59 | 1.55 |
| **Forensic Status** | | | | | |
| NCR-related status | 3564 (48.5) | 81 (28.3) | Ref | | |
| Fitness-related status | 3317 (45.1) | 191 (66.8) | 2.53*** | 1.94 | 3.30 |
| Other reason | 468 (6.4) | 14 (4.9) | 1.32 | 0.74 | 2.34 |
| **Elopement Attempts/Threats** | | | | | |
| Did not exhibit behavior | 7010 (95.4) | 238 (83.2) | Ref | | |
| Exhibited behavior | 339 (4.6) | 48 (16.8) | 4.17*** | 3.00 | 5.79 |
| **Insight into Mental Illness** | | | | | |
| Full or limited insight | 5453 (74.2) | 140 (49) | Ref | | |
| No insight | 1896 (25.8) | 146 (51) | 3.00*** | 2.37 | 3.80 |
| **Medication Adherence** | | | | | |
| ≥80% adherent | 4635 (63.1) | 113 (39.5) | Ref | | |
| <80% adherent | 2714 (36.9) | 173 (60.5) | 2.62*** | 2.05 | 3.33 |
| **Aggressive Behavior Scale, M (SD)** | 1.17 (2.61) | 5.56 (3.49) | 1.40*** | 1.35 | 1.45 |
| **Violence Sum, M (SD)** | 4.86 (3.36) | 8.23 (4.20) | 1.27*** | 1.23 | 1.31 |
| **Mania Scale, M (SD)** | 3.18 (4.09) | 7.24 (4.91) | 1.18*** | 1.53 | 1.20 |
| **Cognitive Performance Scale, M (SD)** | 0.59 (1,01) | 1.19 (1.55) | 1.45*** | 1.33 | 1.57 |

*p < .05, **p < .01, ***p < .001; *OR*, Odd Ratio; CI, confidence interval.

The overall logistic GLMM model for restraint (Table 4) was significant, $F(19, 7609) = 24.177$, $p < .001$, indicating that the predictors significantly explained variability in seclusion. As expected, age and measures of aggression and violence were significantly associated with higher odds of restraint, whereby younger patients and patients exhibiting higher scores on the ABS and Violence Sum scales were more likely to be associated with restraint, partially supporting our first hypothesis for restraint predictors. Regarding diagnoses, mood and anxiety disorders, neurodevelopmental disorder, and personality disorder were significantly associated with restraint. In contrast, other diagnoses, including psychotic disorder and substance use disorder, were not, in partial support of our hypothesis. Compared to being admitted for NCR-related reasons, fitness-related legal status increased the odds of restraint by 44%, and lack of medication adherence and higher scores on our cognitive impairment measure were also significant predictors in this model. However, neither sex, elopement attempts, insight, immigration status, nor Indigeneity were found to be significantly associated with restraint. The ICC was 0.02, indicating a low clustering effect. After adjusting for patient-level predictors, 2% of the variance in restraint was attributable to hospital-level factors (ICC = 0.02), reflecting a small clustering effect. The fixed effects (patient-level predictors) explained 5% of the variance (Marginal pseudo R² = 0.05). When accounting for both patient-level predictors and hospital-level clustering effects, the total explained variance increased to 8% (Conditional pseudo R² = 0.08), indicating that hospital-level clustering contributed additional variance beyond individual-level factors. Table 4 presents the associations between demographic, administrative, and clinical factors and restraint during the first three days of admission with a hospital-level clustering effect.

## Discussion

Few prior studies have addressed individual-level factors contributing to seclusion and restraint in forensic hospitals, especially during the early days of admission. We aimed to confirm the relative contribution of previously identified risk factors associated with seclusion and restraint, on the crucial first three days of admission to forensic hospitals, and to examine the contribution of understudied factors. Among 7635 admissions to forensic hospitals in Ontario between April 2013 and March 2023, 30.2% of patients were secluded, and 3.7% were restrained, similar to findings reported by Hui and colleagues for seclusion [6] and Flammer and colleagues for restraint [3]. Seclusion rates varied over the ten years, with a maximum seclusion rate of 49.7% reported during the first year of the COVID-19 pandemic, consistent with previous reports of an overall increase in patient acuity [61] and coercive measures in psychiatric hospitals and seclusion in particular [62,63].

In multivariate analysis of seclusion accounting for hospital-level clustering effect, younger age, male sex, and measures of violence and aggression were significant contributors, as hypothesized. Previously understudied variables, including Mania, cognitive impairment, and medication non-adherence, added to the predictive model. Furthermore, patients admitted for reasons related to their fitness-related forensic status (i.e., fitness-related orders such as fitness assessment and committal order, treatment order, and keep fit order) were almost twice as likely to be secluded than patients admitted for reasons related to their criminal responsibility at the time of their offense. Similarly, in our adjusted logistic GLMM model of restraint, previously understudied variables related to medication non-adherence, cognitive impairment, and fitness-related status were significant contributors after accounting for younger age and violence and aggression, as well as mood and anxiety, personality, and neurodevelopmental disorder diagnoses.

The present study confirms some previously identified risk factors for coercive measures in forensic psychiatry. For example, young adults were more likely secluded and restrained than older adults, similar to earlier findings [7,42], despite the present sample's mean age being higher than in previous studies [6]. Aggressive and violent behaviors were significantly associated with higher odds of both restraint and seclusion, underscoring research in which actual and threats of violence are cited as reasons for initiating seclusion and restraint events [6,35,40,43]. In addition, being male was a predictive factor for seclusion, adding to evidence from some recent studies [42,43] in contrast to older studies [6]. Sex did not contribute to the odds of being restrained in the adjusted logistic GLMM model. This finding differs from Beghi's

**Table 4. Fixed effects for logistic GLMM model predicting restraint during the first three days of admission with hospital-level clustering effect.**

**Logistic Generalized Linear Mixed Models – GLMM**

| Factor | β | SE | t | p | β2 | 95% C.I. | |
|---|---|---|---|---|---|---|---|
| | | | | | | Lower | Upper |
| **Intercept** | −517 | 0.47 | −10.96 | <.001 | 0.01*** | 0.00 | 0.01 |
| **Age at Admission (years) M** | −0.02 | 0.01 | −4.00 | <.001 | 0.98*** | 0.96 | 0.99 |
| **Sex** | | | | | | | |
| Female | Ref | | | | Ref | | |
| Male | 0.09 | 0.17 | 0.51 | .611 | 1.09 | 0.78 | 1.53 |
| **Diagnoses** | | | | | | | |
| No psychotic disorder | | | | | Ref | | |
| Psychotic Disorder | 0.07 | 0.20 | 0.36 | .721 | 1.07 | 0.73 | 1.58 |
| No Substance Use Disorder | | . | . | . | . | . | . |
| Substance Use Disorder | −0.12 | 0.14 | −0.87 | .382 | 0.88 | 0.67 | 1.16 |
| No Mood and Anxiety Disorder | | . | . | . | . | . | . |
| Mood and Anxiety Disorder | 0.48 | 0.21 | 2.35 | .019 | 1.62* | 1.08 | 2.42 |
| No Neurocognitive Disorder | | . | . | . | . | . | . |
| Neurocognitive Disorder | 0.18 | 0.37 | 0.47 | 0.636 | 1.19 | 0.57 | 2.49 |
| No Personality Disorder | | . | . | . | . | . | . |
| Personality Disorder | 0.34 | 0.17 | 2.00 | .045 | 1.41* | 1.01 | 1.96 |
| No Neurodevelopmental Disorder | | . | . | . | . | . | . |
| Neurodevelopmental Disorder | 0.47 | 0.24 | 1.98 | .048 | 1.60* | 1.00 | 2.55 |
| **Immigration Background** | | | | | | | |
| No immigration background | | | | | Ref | | |
| Has immigration background | 0.18 | 0.17 | 1.06 | .290 | 1.20 | 0.86 | 1.68 |
| **Indigenous Self-Identification** | | | | | | | |
| Did not identify as Indigenous | | | | | Ref | | |
| Identified as Indigenous | 0.01 | 0.29 | 0.04 | .968 | 1.01 | 0.57 | 1.80 |
| **Forensic Status** | | | | | | | |
| NCR-related status | | | | | Ref | | |
| Fitness-related status | 0.37 | 0.16 | 2.24 | .025 | 1.44* | 1.05 | 1.98 |
| Other reason | 0.05 | 0.33 | 0.14 | .889 | 1.05 | 0.55 | 1.99 |
| **Elopement Attempts/Threats** | | | | | | | |
| Did not display behavior | | | | | Ref | | |
| Displayed elopement behavior | 0.33 | 0.20 | 1.66 | .098 | 1.39 | 0.94 | 2.06 |
| **Insight into Mental Illness** | | | | | | | |
| Full or limited insight | | | | | Ref | | |
| No insight | 0.00 | 0.15 | −0.02 | .985 | 1.00 | 0.74 | 1.34 |
| **Medication Adherence** | | | | | | | |
| ≥80% medication adherent | | | | | Ref | | |
| <80% adherent | 0.32 | 0.14 | 2.27 | .023 | 1.38* | 1.04 | 1.82 |
| **Clinical Scales** | | | | | | | |
| Aggressive Behavior Scale | 0.23 | 0.03 | 8.82 | <.001 | 1.26*** | 1.20 | 1.33 |
| Violence Sum | 0.13 | 0.02 | 6.46 | <.001 | 1.14*** | 1.09 | 1.18 |
| Mania Scale | 0.03 | 0.02 | 1.79 | .074 | 1.03 | 1.00 | 1.06 |
| Cognitive Performance Scale | 0.17 | 0.05 | 3.25 | .001 | 1.19** | 1.07 | 1.31 |

*p < .05, **p < .001, ***p < .001; β = coefficient, SE = Standard error, t = t-test*t*, β² = (exp)coefficient, CI = Confidence interval.

and colleagues' findings for general psychiatric patients in which males were more likely than females to be restrained [35]. This difference could be attributable to the small percentage of women in forensic psychiatry (16%) compared to an almost 50:50 split in general psychiatry and only 58 female patients being restrained in the present study, less than 1% of the sample.

The present study offers new insights into previously understudied or equivocal demographic, administrative, and clinical factors associated with seclusion and restraint within the first three days of admission to forensic hospitals. Similar risk factors were identified for both seclusion and restraint. Having an immigration background was not a predictor of seclusion or restraint in our adjusted logistic GLMM models. This result differs from Hansen and colleagues (2020) and Hui and colleagues (2015), who found non-significant positive associations with race and ethnicity in their reviews [6,43]. However, the variable in our dataset included any immigration to Canada at any time in the patient's life, and might not represent racial or ethnic minority individuals. We also examined Indigenous self-identification and found it, too, was not associated with coercive measures in either of the models. This finding contributes to the ongoing discussion about the differences in care between forensic hospitals and correctional institutions, where individuals identifying as Indigenous are at higher risk of seclusion [50,64].

Patients admitted due to fitness-related reasons, such as assessment or due to warrant of committal, treatment, or keep fit orders, were 50 percent more likely secluded or restrained than patients admitted due to NCR-related orders after accounting for all other factors, including hospital-level clustering effects. The higher risk of seclusion and restraint for fitness-related admissions may be because these patients are more likely to be acutely unwell. This is the first study to look at the association of admission status in forensic hospitals with coercive control measures. It is noteworthy that many patients had not yet been tried for their offenses yet were exposed to coercive measures while in hospital.

Elopement behavior significantly decreased the risk of seclusion in the adjusted logistic GLMM after accounting for other factors, despite increasing the odds of seclusion in the unadjusted bivariate analysis and contrary to previous research on coercive measures in general psychiatric settings [35,40]. The adjusted model accounted for aggression, violence, mania, and cognitive impairment, all of which were strong predictors of seclusion. This suggests that when these higher-risk behaviors are present, they may overshadow elopement as a trigger for seclusion, or that elopement may co-occur with less severe clinical presentations. Additionally, staff may respond to elopement attempts with enhanced observation or environmental controls rather than seclusion, particularly in settings where escape is unlikely due to secure infrastructure. It is also possible that elopement behavior in this context reflects a form of goal-directed, non-aggressive behavior that staff do not perceive as warranting seclusion, especially if it is not accompanied by violence or acute agitation. Finally, the temporal focus on the first three days of admission may capture a period when staff are particularly vigilant and may use alternative strategies to manage risk, further reducing reliance on seclusion for elopement attempts. Thus, the inverse association observed in the adjusted model likely reflects a complex interplay between patient behavior, staff response, and the secure forensic environment, rather than a simple causal relationship.

Diagnosis of neurocognitive disorder was negatively associated with seclusion, and neurodevelopmental disorder, personality disorder and mood and anxiety disorder were associated with increased odds of restraint. The association of mood and anxiety disorder with restraint may be attributable to the presence of bipolar disorder, which is a previously well-established risk factor for coercive measures in general psychiatry [36,65,66]. De-Oliveira reported in their multivariable analysis of factors associated with control measures among female and male psychiatric patients that both neurocognitive and neurodevelopmental disorders increased the risk of coercive measures two-fold for women and only neurocognitive diagnosis for men [40]. These associations have not previously been reported for forensic patients. We found that lack of insight and medication non-compliance were significant predictors of seclusion, confirming initial findings by Mathias & Hirdes [7]; however, only medication adherence was a predictor in the adjusted GLMM model for restraint.

We found a moderate clustering effect of forensic facilities in the adjusted logistic GLMM model for seclusion and a minimal effect in the restraint model. The effect on the seclusion model was not clearly attributable to a single outlier,

as one facility (Facility H) was substantially more likely to use seclusion than other hospitals, while another facility used seclusion in the near absence of restraint (Facility D), and another used an atypically high proportion of restraint to seclusion (Facility A). Differences may be due to the level of security within each forensic service, which in our study ranged from minimum security with unlocked units to a high secure building with restricted internal movement, and could affect the need for additional restrictive measures. Patient populations within the hospitals may differ in their level of psychiatric acuity and aggressive behavior, as well as demographic characteristics and legal status, which in turn is related to individual risk for seclusion and restraint. For example, in the Canadian context, some forensic units are designed for persons undergoing assessment and may provide services for a higher proportion of individuals being assessed for their current fitness to stand trial, who are more likely to present with active symptoms of mental illness. Other units serve longer-stay patients found not criminally responsible for offenses and who may be successfully treated for mental illness but remain in the forensic system due to concerns about their risk of reoffending if released to the community. While system-level changes may be needed to implement seclusion and restraint reduction strategies across hospitals, our purpose in the present study was to examine individual-level risk factors that lead to improvements in risk assessment and implementation of individualized interventions to prevent seclusion and restraint.

## Limitations

We conducted a secondary analysis of anonymized health records using each variable's available data and existing operational definitions. For example, we were unable to identify diagnoses of bipolar disorder specifically in many cases; however, we mitigated this by including a measure of manic state, an important risk factor for aggression that may be related to restraint and seclusion. Further, analysis of archival data can have variable reliability and data quality; future studies using prospective clinical assessment are needed to replicate the present findings. Some methodological decisions were necessary during the data management; for example, we excluded multiple admissions each year, enabling our analysis to focus on cases but could have inflated our estimates of coercive measures if at-risk individuals were secluded or restrained upon new admissions in subsequent years. Our data pertained to seclusion and restraint within the first three days of admission, which enabled our examination of risk factors that clinicians can assess upon a patient's admission to a forensic hospital but did not permit an assessment of factors associated with prolonged seclusion. We combined mechanical and physical restraint due to low proportions, even though they may be different strategies and experiences. We acknowledge that our definitions of coercive and restrictive practices may affect prevalence rates and that variability in definitions and reporting in the literature could influence the interpretation of risk factors for such practices. For example, we included only physical restraint (hands-on contact) or mechanical restraint (use of physical devices to restrict movement) in our definition of restraint, whereas Belayneh and colleagues included chemical restraint (medications intended to inhibit an individual's movement or behavior) [2]. Chemical restraint was excluded in our study due to challenges in defining and identifying its use. PRN medications may serve multiple purposes, and clinical records often lack clear documentation of their intended use. Definitions also vary widely across different settings, which limits standardization. To ensure consistency, we employed a more conservative definition that focuses on physical and mechanical restraint. Nonetheless, certain forms of restraint may be systematically more likely to be applied to patients with specific characteristics. For example, Nicholls and colleagues reported that men were more likely to receive PRN antipsychotic medication than women, who were more likely to be placed in seclusion [67]. Our exclusion of chemical restraint may therefore affect our finding that gender was not a risk factor for restraint. Future research using time spent in seclusion or different kinds of restraint is required to determine whether the risk factors may differ.

## Implications

The present study clarifies questions in the literature about whether certain risk factors are consistently associated with coercive measures in forensic settings, such as younger age, aggression, and violent behavior, which were consistent

predictors of both seclusion and restraint in the present study. In addition, understudied indicators, including mania, poor cognitive performance, lack of insight, and medication non-adherence, were consistent predictors. Some demographic characteristics also contributed to both models, even accounting for aggression and mental disorder, indicating the need for further examination of the mechanisms of these associations. For example, staff may perceive young men to be more likely to engage in violent behavior and enact seclusion or restraint to prevent potential harm; different research methods, such as studies of staff perceptions of risk and functional analysis [68], may help elucidate some of the mechanisms.

The present findings provide some evidence to support the practice of individual-level assessments and individualized interventions to reduce the risk of seclusion and restraint. Understanding the factors associated with restraint and seclusion within the first three days of forensic admission allows clinicians to identify risk factors in the crucial time period shortly after a patient is admitted and provide timely interventions to limit the onset of the use of coercive measures. Our findings regarding clinical risk factors suggest valuable interventions among at-risk forensic in-patients including evidence-based pharmacological and psychosocial treatments. In particular, evidence-based interventions for the prevention of in-hospital aggression are indicated by our findings and may include both violence risk assessment [69] and assessments specifically to identify individuals at risk of experiencing coercive measures [5], along with aggression management training programs for clinical staff [70], especially in units caring for young men and individuals under fitness-related assessment or treatment orders.

## Conclusion

In this study of coercive measures during the first three days of admission to forensic hospitals, seclusion was more widespread than restraint, aligning with previous reports. Clinical measures and demographic indicators associated with seclusion and restraint are available to clinicians when forensic patients are admitted to forensic hospitals or shortly afterward. Effectively assessing risk and targeting those needs at admission would reduce the implementation of these coercive measures. Early interventions will benefit patients' rights and well-being and reduce the adverse impact on patients and staff.

## Supporting information

**S1 Table. Demographic, administrative, and clinical characteristics among forensic patients admitted between April 1, 2013 to March 31, 2023.**
(DOCX)

## Author contributions

**Conceptualization:** Elke Ham, Soyeon Kim, N. Zoe Hilton.

**Data curation:** Elke Ham, Soyeon Kim.

**Formal analysis:** Elke Ham.

**Investigation:** Elke Ham.

**Methodology:** Elke Ham, Soyeon Kim.

**Writing – original draft:** Elke Ham.

**Writing – review & editing:** Soyeon Kim, N. Zoe Hilton.

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
