## [Decision Letter · Decision Letter 0]

Dear Dr.  Hilton,

Thank you for submitting your manuscript to PLOS ONE. After careful consideration, we feel that it has merit but does not fully meet PLOS ONE’s publication criteria as it currently stands. Therefore, we invite you to submit a revised version of the manuscript that addresses the points raised during the review process.

We look forward to receiving your revised manuscript.

Kind regards,

Zelalem Belayneh Muluneh

Academic Editor

PLOS ONE

Reviewers' comments:

Reviewer's Responses to Questions

**Comments to the Author**

1. Is the manuscript technically sound, and do the data support the conclusions?

Reviewer #1: Partly

Reviewer #2: Partly

2. Has the statistical analysis been performed appropriately and rigorously?

Reviewer #1: No

Reviewer #2: Yes

3. Have the authors made all data underlying the findings in their manuscript fully available?

Reviewer #1: No

Reviewer #2: No

4. Is the manuscript presented in an intelligible fashion and written in standard English?

Reviewer #1: Yes

Reviewer #2: Yes

Reviewer #1: This is an extremely important and highly relevant topic. As the authors have noted, the use of seclusion and restraints in psychiatric settings raises significant concerns regarding the protection of patient rights and the trauma experienced by patients and staff involved in these measures. The longitudinal data used in this analysis is excellent for examining these issues in forensic mental healthcare.

I found the article engaging and believe it contributes valuable insights, and I would like to see it published. However, several major revisions are needed prior to publication. Broadly, these revisions include providing a more detailed background and discussion of the literature. This should include a deeper analysis of why policies have been unsuccessful in reducing restraints, and a clearer description of the Canadian forensic psychiatric system for readers unfamiliar with it. There are also significant concerns with the methodological approach, particularly the failure to account for clustering effects given the data's collection from multiple hospitals. Lastly, there are some grammatical and stylistic concerns that should be addressed.

1. Abstract and Introduction

The authors clearly summarize the main research question, identifying factors related to seclusion and restraints in forensic mental health care settings. The authors cite relevant literature, particularly highlighting previous work on coercive measures in psychiatric settings and forensic hospitals. However, the paper would benefit from a deeper discussion. Please consider incorporating an overview of the Canadian forensic hospital system and how individuals enter into the healthcare system. Discuss the rationale for restraints and seclusion, and why policies and practices to reduce restraints have not been uniformly successful. Are these failures system-related issues, staffing problems, patient mix/presentation, environmental challenges, etc.?

2. Methods

Selection of method:

More information on hospital-related factors would provide important context. A major methodological concern is that the data was collected from 10 different forensic psychiatric hospitals in Ontario, yet the authors do not account for the potential clustering of patients within these hospitals. Hospital-related characteristics could significantly influence the use of seclusion and restraints. Did the authors consider alternative statistical methods to account for clustering (e.g., GEE)? These models would account for both patient-level and hospital-level characteristics.

Missing data:

The authors do not provide a rationale for why missingness was not considered a concern. This should be addressed.

Psychiatric Diagnoses and Substance Abuse:

Substance abuse often co-occurs with other psychiatric conditions and can exacerbate aggressive behaviours, potentially increasing the likelihood of seclusion or restraint. Could the authors discuss the rationale for using only a primary diagnosis and not accounting for dual diagnosis? There is also no discussion of whether patients are screened for substances at admission which could affect their behaviour. If this information was unavailable in the data, the authors should indicate this in the paper.

4. Results

The figures and tables are generally clear and readable, though there are some discrepancies between the text and the tables, such as missing numbers and percentages in the text (n=%) or the layout was inconsistent.

5. Discussion and Conclusions

The study would benefit from a more detailed discussion of hospital characteristics that could influence the use of coercive measures. Additional context is needed to fully understand the Canadian system, particularly regarding fitness-level factors and how these translate to the wider context. I would also appreciate more discussion in the implications section about how to integrate these findings into practice. For instance, pharmacological treatments are an option, as noted, but this is a population that may be reluctant to take medication. A deeper discussion into possible interventions and their challenges would be valuable.

Reviewer #2: The manuscript presents the results of a study investigating retrospectively risk factors for seclusion and restraint in forensic settings. The authors analyzed retrospectively a large sample of forensic admissions and found specific risk factors for both seclusion and restraint, mostly confirming existing data in general psychiatric settings, with added forensic specificities such as legal admission status.

General comment:

The study addresses an interesting and understudied topic. Issue of coercion in psychiatric forensic institutions is rarely investigated. However, some methodological issues greatly impair the quality of this work.

Introduction:

The introduction is too long. The writing style is very repetitive and thus difficult to read. Even with this long text, the authors fail to point to the specificities of the forensic context compared to general psychiatric settings.

The project’s aims, even if described, are not sufficiently justified. It is unclear why the authors chose this two-step analysis design.

Methods:

The definition of restraint as both mechanical and manual restraint is questionable, as both kinds of restraint generally have different durations and are experienced very differently by patients.

The exact content of the RAI-MH remains very obscure throughout the Methods section. It seems to comprise among other instruments unstandardized scales used to assess clinical characteristics such as insight or compliance. There is no clear explanation as to the content of the RAI-MH or the evaluation process. All used scales must be clearly described with appropriate references, and their items listed.

The authors do not explain why they chose to categorize almost all of the independent variables, such as age, which clearly impairs the quality of the analysis. The categorization of diagnoses, although following international classifications, mixes bipolar disorders and other mood disorders, knowing that maniac states clearly represent an important risk factor for coercive measures.

As to the analysis, authors should again clearly state the hypotheses instead of using numbers. The description of the multivariate logistic regression is insufficient. Authors should justify their decision to use two separate models and clearly describe all variables used in the models and the used method to enter independent variables in the model.

Results:

The Results section is, as the introduction, too long and difficult to read. There are way too many duplicates between text and tables.

The authors keep comparing results of the bivariate analysis and of the multivariate regression, whereas only the multivariate analysis should be extensively considered for discussion. There is no clear added value of using both regression models, as this renders the interpretation of results very confuse.

Discussion:

As in the Results section, the authors systematically compare bivariate and multivariate analysis, which does not add any substantial information for the understanding of the study results. They also fail to clearly provide thorough explanation of the observed risk factors, and to draw clinical consequences of the analysis. The specificities of the forensic setting could be more clearly describe. The different legal statuses for example are difficult to understand for the non-expert reader.

The limitations are insufficiently described and should include some of the elements pointed above.

The implications are too vague and should also include clear clinical or organizational recommendations.

**Do you want your identity to be public for this peer review?** For information about this choice, including consent withdrawal, please see our Privacy Policy

Reviewer #1: No

Reviewer #2: No

---

## [Author Response · Author response to Decision Letter 1]

17 Mar 2025

Dear Dr. Muluneh,

We would like to express our gratitude to you and the reviewers for the time spent reviewing our paper and providing insightful feedback. Your comments have been invaluable and have greatly contributed to the improvements made in the current version. The authors have thoroughly considered each comment and have made every effort to address them all. We have addressed the lack of availability of the raw data and provided the reason why the data are not legally available. We improved the clarity of our aim, firmed up the introduction and expanded on the Canadian forensic mental health system. We expanded the methodology section by further including references for the variables used and utilized an alternative analysis approach as suggested. We have now employed a multilevel logistic regression model using a LOGIT link (Generalized Linear Mixed Model – GLMM) to account for the clustering of responses within the 10 hospitals; we hope that the revised manuscript meets your high standards.

We have uploaded a Response to reviewers document that details our point-by-point responses in a table.

---

## [Decision Letter · Decision Letter 1]

Dear Dr. Hilton,

Thank you for submitting your manuscript to PLOS ONE. After careful consideration, we feel that it has merit but does not fully meet PLOS ONE’s publication criteria as it currently stands. Therefore, we invite you to submit a revised version of the manuscript that addresses the points raised during the review process.

We look forward to receiving your revised manuscript.

Kind regards,

Zelalem Belayneh

Academic Editor

PLOS ONE

Journal Requirements:

Additional Editor Comments:

**Dear Authors,** <o:p></o:p>

Thank you for the opportunity to act as the handling editor for this interesting study.<o:p></o:p>

I have one comment regarding the use of secondary data sources. Existing literature documents inconsistencies in the terminologies and conceptual boundaries used to define coercive or restrictive practices e.g., (https://www.sciencedirect.com/science/article/abs/pii/S1359178924000168 ) and (file:///C:/Users/Zmul0002/Downloads/s00127-024-02739-6%20(4).pdf ). <o:p></o:p>

These definitional inconsistencies may lead to measurement bias, as interpretations of what constitutes a coercive can vary based on clinicians’ subjective judgments (https://onlinelibrary.wiley.com/doi/10.1111/jocn.17588 ). For example, Savage et al (https://pubmed.ncbi.nlm.nih.gov/38205597/ ) noted that variation in the types and numbers of practices reported across studies can significantly impact the reported prevalence rates of coercive practices Such inconsistencies could further confound the measurement outcomes given that you have utilized secondary data sources.<o:p></o:p>

I suggest that you address these issues in the discussion section, particularly under limitations, to acknowledge how definitional and reporting variability might affect the interpretation of your findings.<o:p></o:p>

Reviewers' comments:

Reviewer's Responses to Questions

**Comments to the Author**

Reviewer #2: All comments have been addressed

2. Is the manuscript technically sound, and do the data support the conclusions?

Reviewer #2: Yes

3. Has the statistical analysis been performed appropriately and rigorously?

Reviewer #2: Yes

4. Have the authors made all data underlying the findings in their manuscript fully available?

Reviewer #2: No

5. Is the manuscript presented in an intelligible fashion and written in standard English?

Reviewer #2: Yes

Reviewer #2: The authors have made substantial changes to the manuscript. The readability has been improved. The new methodological approach has also improved the quality of the analysis and the results. The introduction is way more informative than before, and the discussion now addresses most of the relevant findings.

Minor revisions:

- Could the authors give information about the number of incomplete datasets that were excluded from analysis?

- The aspect of elopement behaviour could be discussed some more, as the findings might seem surprising from a general psychiatric point of view.

**Do you want your identity to be public for this peer review?** For information about this choice, including consent withdrawal, please see our Privacy Policy

Reviewer #2: No

---

## [Author Response · Author response to Decision Letter 2]

6 Jun 2025

Zelalem Belayneh

Academic Editor

PLOS ONE

June 6, 2025

Re: revision for PONE-D-24-34524R1

Factors associated with seclusion and restraint on admission to forensic psychiatric hospitals: A 10-year retrospective study

Dear Dr. Zelalem Belayneh,

We sincerely thank you and the reviewers for your time and insightful feedback on our revised paper. Your comments have been invaluable in improving the current version, and we have made every effort to address them. We have clarified the terminologies and conceptual boundaries of our definitions in the limitations section, responded to the reviewers' concerns regarding missing data, and provided further elaboration on the findings related to elopement in the adjusted GLMM model in the discussion.

Please see our point-by-point responses in the table below.

Comments: Editor Response 2nd revision

I have one comment regarding the use of secondary data sources. Existing literature documents inconsistencies in the terminologies and conceptual boundaries used to define coercive or restrictive practices e.g., (https://www.sciencedirect.com/science/article/abs/pii/S1359178924000168 ) and (file:///C:/Users/Zmul0002/Downloads/s00127-024-02739-6%20(4).pdf ).

These definitional inconsistencies may lead to measurement bias, as interpretations of what constitutes a coercive can vary based on clinicians’ subjective judgments (https://onlinelibrary.wiley.com/doi/10.1111/jocn.17588 ). For example, Savage et al (https://pubmed.ncbi.nlm.nih.gov/38205597/ ) noted that variation in the types and numbers of practices reported across studies can significantly impact the reported prevalence rates of coercive practices Such inconsistencies could further confound the measurement outcomes given that you have utilized secondary data sources.

I suggest that you address these issues in the discussion section, particularly under limitations, to acknowledge how definitional and reporting variability might affect the interpretation of your findings.

RESPONSE - Thank you for bringing this issue regarding inconsistencies in the terminologies and conceptual boundaries to our attention and for your recommendation. We now state in the limitations: We acknowledge that our definitions of coercive and restrictive practices may affect prevalence rates and that variability in definitions and reporting in the literature could influence the interpretation of risk factors for such practices. For example, we included only physical restraint (hands-on contact) or mechanical restraint (use of physical devices to restrict movement) in our definition of restraint, whereas Belayneh et al., 2024 included chemical restraint (medications intended to inhibit an individual's movement or behavior). Chemical restraint was excluded in our study due to challenges in defining and identifying its use. PRN medications may serve multiple purposes, and clinical records often lack clear documentation of intent. Definitions also vary widely across settings, limiting standardization. To ensure consistency, we used a more conservative definition focused on physical and mechanical restraint. Nonetheless, it is possible that certain forms of restraint are systematically more likely to be applied to patients with specific characteristics. For example, Nicholls et al (2009) reported that men were more likely to receive PRN antipsychotic medication than women, who were more likely to be placed in seclusion. Our exclusion of chemical restraint may therefore affect our finding that gender was not a risk factor for restraint.

Reviewer #2: The authors have made substantial changes to the manuscript. The readability has been improved. The new methodological approach has also improved the quality of the analysis and the results. The introduction is way more informative than before, and the discussion now addresses most of the relevant findings.

Minor revisions:

- Could the authors give information about the number of incomplete datasets that were excluded from analysis?

- The aspect of elopement behaviour could be discussed some more, as the findings might seem surprising from a general psychiatric point of view.

RESPONSE: We are pleased that we were able to improve the readability and quality of analysis and results in our prior revision, and that the introduction and discussion are now more informative and streamlined. Regarding missing data, the dataset was complete with no missing values. As noted in the data sources section, all variables were mandatory reporting requirements for Ontario hospitals and had to be completed within 72 hours of admission to a mental health bed. This ensured a fully complete dataset, with no concerns related to missingness in the analysis.

Regarding your suggestion to elaborate on the aspect of elopement behaviour, the Ontario Mental Health Reporting System Resource Manual, 2017–2018 defines elopement attempts/threats as: The person attempts to leave, or actually leaves, the unit without the staff’s knowledge and/or formal discharge (e.g., unauthorized leave of absence). This includes the intent to abscond without leave for any reason, as well as occasions when the person threatens to leave the unit, making him or her an elopement risk. We now state in the discussion:

The adjusted model accounted for aggression, violence, mania, and cognitive impairment, all of which were strong predictors of seclusion. This suggests that when these higher-risk behaviors are present, they may overshadow elopement as a trigger for seclusion, or that elopement may co-occur with less severe clinical presentations. Additionally, staff may respond to elopement attempts with enhanced observation or environmental controls rather than seclusion, particularly in settings where escape is unlikely due to secure infrastructure. It is also possible that elopement behavior in this context reflects a form of goal-directed, non-aggressive behavior that staff do not perceive as warranting seclusion, especially if it is not accompanied by violence or acute agitation. Finally, the temporal focus on the first three days of admission may capture a period when staff are particularly vigilant and may use alternative strategies to manage risk, further reducing reliance on seclusion for elopement attempts. Thus, the inverse association observed in the adjusted model likely reflects a complex interplay between patient behavior, staff response, and the secure forensic environment, rather than a simple causal relationship.

---

## [Decision Letter · Decision Letter 2]

Factors associated with seclusion and restraint on admission to forensic psychiatric hospitals: A 10-year retrospective study

PONE-D-24-34524R2

Dear Dr. N. Zoe Hilton,

We’re pleased to inform you that your manuscript has been judged scientifically suitable for publication and will be formally accepted for publication once it meets all outstanding technical requirements.

Kind regards,

Zelalem Belayneh

Academic Editor

PLOS ONE

Additional Editor Comments (optional):

Reviewers' comments:

Reviewer's Responses to Questions

**Comments to the Author**

Reviewer #2: All comments have been addressed

2. Is the manuscript technically sound, and do the data support the conclusions?

Reviewer #2: Yes

3. Has the statistical analysis been performed appropriately and rigorously?

Reviewer #2: Yes

4. Have the authors made all data underlying the findings in their manuscript fully available?

Reviewer #2: No

5. Is the manuscript presented in an intelligible fashion and written in standard English?

Reviewer #2: Yes

Reviewer #2: All comments have been addressed. I find the manuscript suitable for publication. Congratulations to the authors.

**Do you want your identity to be public for this peer review?** For information about this choice, including consent withdrawal, please see our Privacy Policy

Reviewer #2: No

---

## [Editor Report · Acceptance letter]

PONE-D-24-34524R2

PLOS ONE

Dear Dr. Hilton,

I'm pleased to inform you that your manuscript has been deemed suitable for publication in PLOS ONE. Congratulations! Your manuscript is now being handed over to our production team.

Kind regards,

on behalf of

Mr. Zelalem Belayneh

Academic Editor

PLOS ONE